# Defense Responses in the Interactions between Medicinal Plants from Lamiaceae Family and the Two-Spotted Spider Mite *Tetranychus urticae* Koch (Acari: Tetranychidae)

**Katarzyna Golan** [1,*], **Inmaculada Garrido Jurado** [2], **Izabela Kot** [1], **Edyta Górska-Drabik** [1], **Katarzyna Kmieć** [1], **Bożena Łagowska** [1], **Barbara Skwaryło-Bednarz** [1], **Marek Kopacki** [1] and **Agnieszka Jamiołkowska** [1]

[1] Department of Plant Protection, University of Life Sciences in Lublin, Leszczyńskiego 7, 20-069 Lublin, Poland; izabela.kot@up.lublin.pl (I.K.); edyta.drabik@up.lublin.pl (E.G.-D.); katarzyna.kmiec@up.lublin.pl (K.K.); bozena.lagowska@up.lublin.pl (B.Ł.); barbara.skwarylo@up.lublin.pl (B.S.-B.); marek.kopacki@up.lublin.pl (M.K.); aguto@wp.pl (A.J.)

[2] Department of Agronomy, University of Cordoba, Campus de Rabanales Building C4, 14071 Cordoba, Spain; g72gajui@uco.es

\* Correspondence: katarzyna.golan@up.lublin.pl

**Abstract:** This study aimed to determine the effects of plant species on the biological parameters of *Tetranychus urticae* Koch and the time of mite infestation on plant physiology in *Ocimum basilicum* L., *Melissa officinalis* L. and *Salvia officinalis* L. Mite infestation induced various levels of oxidative stress depending on plant species and the duration of infestation. Host plants affected *T. urticae* life table parameters. The low level of susceptibility was characteristic of *S. officinalis*, which appeared to be the least infected plant species and reduced mites demographic parameters. Infested leaves *of S. officinalis* contained elevated levels of hydrogen peroxide ($H_2O_2$) and malondialdehyde (MDA) compared to control. In addition, higher membrane lipid peroxidation and higher activity of guaiacol peroxidase (GPX) and lower activity of catalase (CAT) were recorded with a longer mite infestation. In contrast, *O. basilicum* appeared to be a suitable host on which *T. urticae* could develop and increase in number. In basil leaves, increasing levels of hydrogen peroxide and MDA with elevated GPX activity and strongly decreased catalase activity were recorded. Knowledge of the differences in mite susceptibility of the tested medicinal plants described in this study has the potential to be applied in breeding strategies and integrated *T. urticae* pest management in medicinal plant cultivations.

**Keywords:** two-spotted spider mite; plant acceptance; biological parameters; plant physiology; oxidative stress



## 1. Introduction

The production and market of plant material used in health and medical products has been substantially increasing in the last three decades due to increasing demand; in fact, 30% of worldwide sold drugs are based on plants [1,2]. In recent years, the production of herbal plants in containers under cover is one of the rapidly developing branches of greenhouse production all over the world. The amount of pesticides applied in greenhouses during the growing period of medicinal plants is reduced, but similar to other greenhouse crops, those plants are adversely affected by pests and diseases during their development [3–6]. Among various arthropod species, mites are an increasingly important group of phytophagous pests on medicinal plants, especially on greenhouse crops [7–9]. Both mite and insect use plants as food source, causing significant damage and yield losses.

The two-spotted spider mite, *T. urticae* Koch (Acari: Tetranychidae), represents one of the most destructive generalists among mite herbivores. The complete life cycle of *T. urticae* may occur in less than 10 days. Females, during their lifetime, may produce over 100 eggs under optimal conditions. This pest feeds on more than 1100 plant species belonging to various botanical families, 150 of which are important agricultural crops, including herbs

from the family Lamiaceae [9–11]. *T. urticae* feeds on plant tissue by piercing mesophyll cells and introducing a stylet between epidermal cells or through the stomata, injecting saliva to predigest the cell content and suck it up. Consequently, they cause biochemical and physiological changes in plant tissues, disrupt cell physiology, reduce photosynthesis and inject phytotoxic compounds. This results in the appearance of necrotic or yellowing spots and a darkening of the damaged plant organ, which usually turn yellow, gray and consequently fall off [12–17]. The feeding of mites causes a reduction in yield and decrease the quality of plants [18–20]. Due to its polyphagous nature and short life cycle, enormous reproductive capacity and remarkable ability to develop resistance to pesticides, it is considered a serious threat to agriculture, including the production of herbal plants in containers under cover [3,4,21–23].

As documented, the most common properties of plants causing avoidance behaviour of herbivores are related to the presence of trichomes and wax structures on plant surface, leaf thickness and toughness, sclerotization [14,15,24]. Short hairs or trichomes may impede arthropods or prevent them from feeding on a particular plant leaf. Plant physical barriers to *T. urticae* stylet penetration into the leaf mesophyll and various phytochemicals have been identified as potent constitutive defenses that can directly restrict two-spotted spider mite feeding and/or fitness [15].

Herbivores feeding on plant tissues induce a broad range of defense responses, including the generation of reactive oxygen species (ROS) in cells. Oxidative stress usually results from excessive ROS production, which can cause severe oxidative damage to plants [25–27]. ROS comprise molecules such as superoxide, hydrogen peroxide, hydroxyl radicals, and singlet oxygen that play a prominent role in plant response to numerous stresses, including plant interactions with herbivores. ROS may initiate destructive oxidative processes, such as chlorophyll bleaching, lipid peroxidation, protein oxidation, and nucleic acid damage, eventually leading to cell death [16,28,29]. However, plants have antioxidant mechanisms to scavenge excess ROS and prevent cell damages [30–34]. Low molecular weight antioxidants (ascorbic acid, glutathione and tocopherols) and ROS-scavenging enzymes have the capacity to scavenge superoxides, hydroxyl radicals and singlet oxygen [35,36]. These molecules are most commonly activated to protect stressed plant tissue against damage caused by overproduction of potentially harmful ROS [37]. Generally, it seems that the greater the ROS balancing capacity, the higher the stress tolerance [37–41]. As documented, peroxidases catalyze oxidoreduction between $H_2O_2$ and various reductants, such as many phenolic compounds, they participate in the wall-building processes, such as suberization and lignifications, phenol oxidation, auxin catabolism and wound healing, as well as defense against the feeding of insects and mites [42,43]. In turn, catalase directly converts $H_2O_2$ into $N_2O$ and $O_2$ [44]. While ROS molecules are necessary to orchestrate defense responses, their effect on plant-resistance/susceptibility to a particular herbivore is very specific and depends on plant-herbivore interaction [16].

*T. urticae* is considered a model species among phytophagous chelicerates for studying various aspects of plant-spider mite interactions. Its ability to feed on a wide range of plant species has provided a unique opportunity for functional studies of plant-mite interactions [7]. Understanding the mechanisms by which *T. urticae* adapts to different plant defense strategies, may complement the management of this pest [45]. Many aspects of spider mite biology, including rapid development, high fecundity and haplodiploid sex determination, seem to facilitate rapid evolution of pesticide resistance [46]. However, plant defense responses to *T. urticae* constitute a complex and highly regulated process involving many factors, signaling molecules and pathways [45].

Although a great number of studies concerning plant-mite interactions have been conducted in recent years, information about the two-spotted spider mite feeding on medicinal plants is still lacking [45,47]. In addition, little is known about the antioxidant defense system mechanisms under *T. urticae* infestation, as well as the role of $H_2O_2$ in signal transduction against spider mites [17,45].

Therefore, the present work aimed to study plant-mite interactions selecting sweet basil (*Ocimum basilicum* L.), lemon balm (*Melissa officinalis* L.) and sage (*Salvia officinalis* L.) as common medicinal plants in the family Lamiaceae on the basis of their economical relevance and their various level of susceptibility against spider mites in greenhouses. To address this issue, choice and non-choice tests were performed in order to assess the host preference of *T. urticae* and how it affects mite biological parameters (oviposition and fecundity, longevity, life cycle period). Additionally, our attention was focused on the presence of trichomes in the selected plants as local response and the oxidative stress as systemic response by measuring the level of membrane lipid peroxidation and the activity of selected antioxidant enzymes (guaiacol peroxidase (GPX) and catalase (CAT)) in the selected plants at the same developmental stage in response to mite pest and duration of infestation.

## 2. Material and Methods

### 2.1. Plant Material

Three herb species from Lamiaceae family: sweet basil (*Ocimum basilicum* L.), lemon balm (*Melissa officinalis* L.) and sage (*Salvia officinalis* L.) were used for analysis. Herbal seeds were purchased from PNOS S.A. in Ozarów Mazowiecki, Poland.

All plants used in this study were cultivated in a growth chamber (1.8 m × 2.5 m × 2.5 m) with metal walls covered in a white enamel finish. As the source of light Greenie LED T8 Flora lamps were used. The same experimental conditions were applied for all the plants used during all periods of this study (day/night temperature 24/18 ± 1 °C; humidity 60 ± 5%, photoperiod L:D = 16:8). The age of each plant was determined from the day of seeding. The plants were sown in plastic containers with a standardized garden substrate on two separated tables of 2.50 m$^2$ in size. About 7 weeks after sowing, young plants were transplanted into 15-cm-diameter pots (three plants per pot) and placed at a distance of 20 cm from each other. The plants were kept on textile subirrigation mats (Polprotex) covered with black agrofabric. During the experiment, the plants were watered from containers every two or three days. The experiment included fully developed basil, lemon balm and sage (56 d after sowing), with 9–16 branches per plant and about 37–53 cm height, depending on the herb species.

### 2.2. Measurement of Trichomes

To perform light microscopy (LM) analysis, the epidermis layer was removed from fresh leaves and immediately embedded in glycerine. Sections, cut transversally with a razor blade, were subsequently stained with Sudan Red and were observed under a Motic microscope.

### 2.3. Choice Test

The free-choice test was carried out 56 d after sowing to analyze the degree of host plant acceptance by mites. Non-infested *O. basilicum*, *M. officinalis* and *S. officinalis* were arranged in a circle on tables in an air-conditioned chamber, so that their leaves did not touch one another. A few leaves of Phaseolus sp. colonized by 150 females *T. urticae* were taken and placed in the center of the circle, on a cardboard platform. The number of mite individuals moving to the plants of the examined species was counted after 24 h of experiment. The level of plant species acceptance was calculated as a percentage of individuals observed on each plant in relation to the number of all *T. urticae* individuals on all plants. The experiment was established in six replicates.

### 2.4. Non-Choice Test: Mites and Plant Colonization

Three fully developed herbal plants were colonized with *T. urticae* individuals derived from laboratory cultures maintained on bean (*Phaseolus* L.) for two weeks before the experiment. Fifteen *T. urticae* females were transferred onto each herbal plant using a bristle brush. Every sample consisted of five plants of one species, on which two-spotted

spider mites were feeding for 24 h, 7 d and 14 d. Uninfested basil, lemon balm and sage plants served as controls. The experiment was conducted in three replications for each plant species. Freshly collected fully developed leaves from plants of each infestation duration were used for physiological analyses.

In order to determine the degree of plant colonization by spider mites, 15 leaves were collected from each of the plants after 14 days of colonization, and the number of individuals inhabiting them was counted per 1 cm$^2$ of leaf. Leaf damage caused by mite feeding was assessed visually and expressed as a percentage of the leaf area.

### 2.5. Biological Parameters of T. urticae

All the experiments were conducted in a growth chamber under the same experimental conditions as applied for the plants. Leaf discs of the host plants were cut out and placed upside down on a cotton layer in Petri dishes (8 cm diameter × 2 cm height) with an opening in the center of the lid for ventilation. The discs were surrounded with wet cotton to keep them fresh and avoid the escape of mites. For experiments, to obtain eggs of the same age, pairs of female and male mites were randomly selected from the colony feeding on plant species colonized for the experiment and transferred onto each leaf disc of the same host plant species. Mites were allowed to mate and lay eggs for 24 h. Eggs laid by each female were collected separately (1 egg per disc) and were maintained under the same conditions to develop. Whenever a leaf began to deteriorate, it was replaced with a fresh leaf. To gain the information about the duration of *T. urticea* life cycle the developmental time of progeny stages were monitored daily and summarized until the death of the last adult. The number of replicates was 15 for for each tested plant species.

To estimate female longevity, fecundity and the mean duration of oviposition period, one female teleiochrysalids in the last developmental stage before adult emergence and a newly emerged male were transferred onto a new test arena. The males were removed 24 h after the female emerged. Eggs were counted twice daily until female death. The number of replicates was 15 for each tested plant species.

### 2.6. Physiological Analysis

#### 2.6.1. Hydrogen Peroxide Concentration

Hydrogen peroxide concentration was estimated by forming titanium-hydro peroxide complex [48]. Subsequently, 0.5 g of plant material was ground in 3 mL of phosphorus buffer (50 mM, pH 6.5) at 4 °C and then, the mixture was centrifuged at $6000 \times g$ for 25 min. Next, 1.5 mL of the supernatant was added to 0.5 mL $TiO_2$ in 20% (*v*/*v*) $H_2SO_4$ and centrifuged again at $6000 \times g$ for 15 min at room temperature. The absorbance of the supernatant was measured at 410 nm against blank reagent with a Cecil CE 9500 spectrophotometer. The $H_2O_2$ concentration in the sample was calculated using the molar absorbance coefficient, which for $H_2O_2$ was 0.28 $\mu M^{-1}\ cm^{-1}$, and it was expressed in nanomoles per 1 g fresh weight.

#### 2.6.2. Malondialdehyde Content (MDA)

The level of peroxidation of membrane lipids was assessed by determining the content of malondialdehyde (MDA) according to the method by Heath and Packer [49]. The crushed plant material (0.2 g) was extracted in a 0.1 M potassium phosphate buffer pH = 7.0, and then centrifuged at $12,000 \times g$ for 20 min. Next, 0.5 cm$^3$ of the homogenate was added to 2 cm$^3$ 20% trichloroacetic acid (TCA) containing 0.5% thiobarbituric acid (TBA) and incubated for 30 min in a water bath at 95 °C.

After incubation, the samples were quickly cooled down and centrifuged again at $10,000 \times g$ for 10 min. Absorbance was measured at 532 and 600 nm using a Cecil spectrophotometer CE 9500. The concentration of malondialdehyde in a sample was calculated by using the molar absorbance coefficient, which for MDA is 155 $nM^{-1}\cdot cm^{-1}$, and it was expressed in nanomoles in 1 g of dry weight.

### 2.6.3. Determination of the Antioxidant Enzymes Activity

Preparation of Extract

The leaves (0.2 g) were homogenized in a mortar and pestle in a 0.05 M phosphorus buffer, containing 0.2 M Ethylene-Diamine-Tetra-Acetate acid (EDTA) and 2% Poly-Vinyl-Pyrrolidone (PVP) with pH = 7.0 at the temperature of 4 °C. The homogenate was then centrifuged at $10,000 \times g$ for 10 min. at 4 °C. The supernatant obtained in this way was used directly for enzyme analysis. The enzymes activity was measured using Cecil CE 9500 spectrophotometer (Cecil Instruments).

Guaiacol peroxidase (GPX, EC 1.11.1.7) activity was measured by the method of Małolepsza et al. [50]. The reaction mixture contained 0.5 cm$^3$ 0.05 M phosphate buffer, pH 5.6, 0.5 cm$^3$ 0.02 M guaiacol, 0.5 cm$^3$ 0.06 M H$_2$O$_2$ and 0.5 cm$^3$ of enzyme extract. The change in absorbance was measured at 480 nm for 4 min, at 1min intervals using a Cecil CE 9500 spectrophotometer. GPX activity was determined using the absorbance coefficient for this enzyme, which was 26.6 mM cm$^{-1}$ and expressed as the change of peroxidase activity per fresh weight (U mg$^{-1}$ FW).

Catalase (CAT; EC 1.11.1.6) activity was determined as described by Chance and Meahly [51] and modified by Wiloch et al. [52]. The reaction mixture contained 2 cm$^3$ 50 mM K-phosphorus buffer, pH 7.0, 0.2 cm$^3$ H$_2$O$_2$ and 0.1 cm$^3$ enzymatic extract. Extinction was measured for 3 min using a Cecil CE 9500 spectrophotometer reading the initial and final results at 240 nm. Catalase activity was determined using the absorbance coefficient, which for catalase is 0.036 mM cm$^{-1}$. The result was converted to catalase activity per fresh weight, expressed as U $\times$ mg$^{-1}$ fresh weight.

### 2.7. Statistical Analysis

All data were presented as means ($\bar{x}$) with standard errors values ($\pm$SE) and statistical analyses were conducted using Statistica for Windows v. 13.1 [53]. The significance threshold was set at $\alpha \leq 0.05$. The Shapiro–Wilk test was used to assess normality. The results of choice test, the mite biological parameters, the level of acceptance, mean number of leaf trichomes and damage of leaves cause by *T. urticae* as well as concentration of hydrogen peroxide and malondialdehyde and the activity of antioxidant enzymes across the assessment period was analyzed using the Kruskal–Wallis one-way non-parametric AOV analysis, for not fulfilling the assumptions of variance homogeneity and normality. The results of the statistical analysis of the free-choice test and non-choice test to determine a significant difference in analyzed means carried out with the Kruskal-Wallis' test reported with an H statistic, degrees of freedom and the P value are included in the Tables 1 and 2. Physiological assays were performed in three independent replicates. The strength of relationship between two variables was described using Spearman's correlation coefficient (r). The following correlation were calculated during this study: the number of trichomes on leaves and plant acceptation by mites; total fecundity; percent of leaves damages; number of mites/1 cm$^2$ of leaves and plant acceptation—total fecundity; percent of leaves damages.

### 3. Results

### 3.1. Plants Infestation by T. urticae

The choice test showed significant differences in the percentage of *T. urticae* individuals per host plant 24 h after beginning of the experiment (H (2, N = 18) = 15.174; $p = 0.001$). *O. basilicum* was the most attractive host with 50% of mites, while feeding on *S. officinalis* was the lowest, as 21.87% of mites were present on this plant after 24 h of the experiment (Figure 1).

The non-choice test showed that *O. basilicum* was the most abundantly colonized host species by comparison of mean number of mites individuals noted on 1 cm$^2$ leaves (Table 1). The mean number of mites/1 cm$^2$ on sweet basil was near 4-fold higher compared to lemon balm, and over 14-fold higher compared to sage. Similarly, *O. basilicum* was characterized by highest value of leaves damage. This parameter on day fourteen corresponded to

an average leaf area of 47.36% on sweet basil and approximately 17.30% on sage plants (Table 1). Subsequent plant species differed in a mean number of leaf trichomes observed. Their highest number was noted on the leaves of *S. officinalis*, their number on leaves of *M. officinalis* was 1.2–fold lower, while on *O. basilicum* it was as much as 17.4–fold lower than in *S. officinalis* (Table 1).

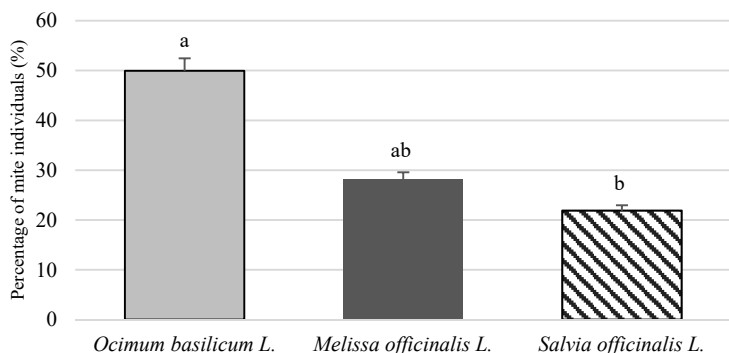

**Figure 1.** Level of *Ocimum basilicum*, *Melisa officinalis* and *Salvia officinalis* acceptance after 24 h of *T. urticae* infestation. Bars marked by different letters are significantly different at $p \leq 0.05$ (Kruskal–Wallis' test).

Different species of medicinal plants significantly affected the tested biological parameters of *T. urticae* (Table 2). Statistical analysis confirmed the occurrence of significantly lower values of oviposition period for two-spotted spider mite females feeding on *O. basilicum* (10.81 days) compared to females on lemon balm and sage, 13.67 days and 14.01 days, respectively. The longevity of female *T. urticae* adults feeding on tested host plants was significantly different and ranged from 13.96 days on sweet basil to 18.34 days on sage plants. Similarly, life cycle period varied between the medicinal plants (H (2, N = 45) = 36.71 P = $1 \times 10^{-14}$) and ranged from 24.30 days on *O. basilicum* to highest value on *S. officinalis* (29.22 days). There were significant differences in the total and daily fecundity rate of *T. urticae* on the tested plants and the highest values were observed on *O. basilicium* (Table 2).

Correlation analysis confirmed the presence of significant strong negative relationships between the degree of plant acceptance and the number of leaf trichomes (r = −0.897; $p = 1 \times 10^{-14}$), total fecundity (as a measure of host quality) and the number of leaf trichomes (r = −0.895; $p = 1 \times 10^{-14}$), total female fecundity and percentage of leaf damage (r = −0.894; $p = 1 \times 10^{-14}$), the number of leaf trichomes and number of mites/1 cm$^2$ (r = −0.728; $p = 1 \times 10^{-14}$). The degree of plant acceptance was positively correlated with total fecundity of *T. urticae* females (r = 0.930; $p = 1 \times 10^{-14}$).

**Table 1.** The level of acceptance of medicinal plants by *Tetranychus urticae* and morphological characteristics of plants leaves with percent of leaves damages caused by mite feeding.

| Host Plant | Number of Mites/1 cm$^2$ of Leaves Mean (SE) (%) | Damage of Leaves after 14 Days Feeding Mean (SE) (%) | Number of Leaf Trichomes/1 mm$^2$ Mean (SE) |
|---|---|---|---|
| *Ocimum basilicum* L. | 3.98 (±0.18) a | 47.36 (±0.49) a | 5.17 (±0.02) c |
| *Melissa officinalis* L. | 1.02 (±0.048) b | 39.53 (±0.29) b | 72.99 (±0.13) b |
| *Salvia officinalis* L. | 0.28 (±0.02) c | 17.30 (±0.15) c | 89.90 (±0.38) a |
| H (2, N = 45) | 39.13 | 39.13 | 39.13 |
| P | $1 \times 10^{-14}$ | $1 \times 10^{-14}$ | $1 \times 10^{-14}$ |

Values in the same column marked by different letters indicate statistically significant differences at $p \leq 0.01$ (Kruskal–Wallis' test).

**Table 2.** Mean (±SE) biological parameters of *Tetranychus urticae* females reared on different species of medicinal plants.

| Plant Species | Periods in Days | | | Total Fecundity | Daily Rate |
| | Oviposition Period | Female Adult Longevity | Life Cycle | | |
|---|---|---|---|---|---|
| *Ocimum basilicum* L. | 10.81 (±0.12) b | 13.96 (±0.12) c | 24.30 (±0.20) c | 55.81 (±0.13) a | 5.16 (±0.08) a |
| *Melissa officinalis* L. | 13.67 (±0.20) a | 17.20 (±0.12) b | 26.43 (±0.65) b | 43.18 (±0.21) b | 3.16 (±0.05) b |
| *Salvia officinalis* L. | 14.01 (±0.13) a | 18.34 (±0.08) a | 29.22 (±0.53) a | 29.12 (±0.03) c | 2.08 (±0.12) c |
| Kruskal Wallis' test H (2, N = 45) P | 29.80 $1 \times 10^{-14}$ | 38.45 $1 \times 10^{-14}$ | 36.71 $1 \times 10^{-14}$ | 36.70 $1 \times 10^{-14}$ | 39.15 $1 \times 10^{-14}$ |

Values in the same column marked by different letters indicate statistically significant differences at $p \leq 0.01$ (Kruskal–Wallis' test).

### 3.2. Changes in Hydrogen Peroxide Concentration

Significant changes in hydrogen peroxide concentration were found between host plants (H = 97.07; $p < 0.001$), but not between consecutive days of the experiment (H = 0.75; $p = 0.53$). $H_2O_2$ concentration ranged from 49.36 for *O. basilicum* leaves, 7 days after treatment to 2.52 nmol $g^{-1}$ FW for control leaves of *M. officinalis*. Higher $H_2O_2$ levels were obtained in *O. basilicum* leaves, and they were significantly higher in infested plants after 7 days of spider mite feeding compared to control plants (Table 3).

**Table 3.** Hydrogen peroxide concentration (nmol $g^{-1}$ FW) within leaves of *Ocimum basilicum*, *Melissa officinalis* and *Salvia officinalis* after *Tetranychus urticae* infestation.

| Treatments [1] | Host Plant [2] | | |
| | *O. basilicum* | *M. officinalis* | *S. officinalis* |
|---|---|---|---|
| Control | 8.86 (±0.01) aA | 2.52 (±0.25) aB | 3.33 (±0.17) aAB |
| 1 day | 36.14 (±0.02) abA | 3.26 (±0.13) aB | 6.05 (±0.45) aAB |
| 7 days | 49.36(±0.01) bA | 2.71 (±0.19) aB | 3.67 (±0.16) aAB |
| 14 days | 29.64 (±0.02) abA | 2.83 (±0.03) aB | 6.81 (±0.50) aAB |

[1] Treatments are as follow: control (leaves of plants without mites), 1 day (leaves of plants after 1 day of infestation), 7 days (leaves of plants after 7 days of infestation) and 14 days (leaves of plants after 14 days of infestation). [2] Means (±SE) with the same letter within columns (lowercase letters) and within rows (uppercase letters) are not significantly different ($p \leq 0.05$; Kruskal–Wallis' test).

### 3.3. Changes in Malondialdehyde Concentration

Malondialdehyde concentration at different periods of mite infestation differed significantly between host plants (H = 9.31; $p < 0.001$) and between successive days of the experiment (H = 10.05; $p < 0.001$). MDA concentration ranged between 15.81 and 3.15 nmol $g^{-1}$ FW for *O. basilicum* leaves and control leaves of *S. officinalis*, respectively, 7 days after treatment. As for $H_2O_2$ concentration, higher MDA level were obtained in *O. basilicum* leaves, and its value measured after 7 days of infestation were significantly higher than in control plants (Table 4).

**Table 4.** Malondialdehyde concentration (nmol $g^{-1}$ FW) within leaves of *Ocimum basilicum*, *Melissa officinalis* and *Salvia officinalis* after *Tetranychus urticae* infestation.

| Treatments [1] | Host Plant [2] | | |
| | *O. basilicum* | *M. officinalis* | *S. officinalis* |
|---|---|---|---|
| Control | 4.03 (±0.00) bA | 3.25 (±0.16) bA | 3.15 (±0.14) bA |
| 1 day | 14.84 (±0.02) abA | 3.87 (±0.12) abB | 4.57 (±0.21) abAB |
| 7 days | 15.81 (±0.00) aA | 4.08 (±0.21) abB | 4.89 (±0.14) abAB |
| 14 days | 13.74 (±0.02) abA | 5.89 (±0.05) aAB | 5.47 (±0.34) aB |

[1] Treatments are as follow: control (leaves of plants without mites), 1 day (leaves of plants after 1 day of infestation), 7 days (leaves of plants after 7 days of infestation) and 14 days (leaves of plants after 14 days of infestation). [2] Means (±SE) with the same letter within columns (lowercase letters) and within rows (uppercase letters) are not significantly different ($p \leq 0.05$; Kruskal–Wallis' test).

### 3.4. Changes in Antioxidant Enzymes Activity

Statistical analysis showed significant differences in guaiacol peroxidase activity between host plants (H = 13.47; $p < 0.001$) and consecutive days of the experiment (H = 7.10; $p < 0.001$). GPX values ranged between 0.29 and 58.38 U mg$^{-1}$ FW for control leaves of *M. officinalis* and *S. officinalis* leaves 1 day after treatment, respectively. Higher values of GPX activity were obtained in *S. officinalis* leaves from infested plants (Table 5). GPX activity in *S. officinalis* leaves was decreasing over time after infestation; in contrast, GPX activity measured in *O. basilicum* leaves was increasing over time (Table 5).

**Table 5.** Guaiacol peroxidase activity (U mg$^{-1}$ FW) within leaves of *Ocimum basilicum*, *Melissa officinalis* and *Salvia officinalis* after *Tetranychus urticae* infestation.

| Treatments [1] | Guaiacol Peroxidase Activity (GPX) [2] | | |
| --- | --- | --- | --- |
| | *O. basilicum* | *M. officinalis* | *S. officinalis* |
| Control | 1.62 ($\pm$0.01) bAB | 0.29 ($\pm$0.05) bA | 4.14 ($\pm$0.19) bB |
| 1 day | 3.73 ($\pm$0.02) abA | 4.09 ($\pm$0.56) abA | 58.38 ($\pm$3.74) aA |
| 7 days | 6.16 ($\pm$0.02) abB | 6.74 ($\pm$0.34) aB | 54.24 ($\pm$1.84) abA |
| 14 days | 9.98 ($\pm$0.00) aAB | 2.96 ($\pm$0.03) abA | 27.07 ($\pm$1.42) abA |

[1] Treatments are as follow: control (leaves of plants without mites). 1 day (leaves of plants after 1 day of infestation). 7 days (leaves of plants after 7 days of infestation) and 14 days (leaves of plants after 14 days of infestation). [2] Means ($\pm$SE) with the same letter within columns (lowercase letters) and within rows (uppercase letters) are not significantly different ($p \leq 0.05$; Kruskal–Wallis' test).

Catalase activity showed significant differences between host plants (H = 19.02; $p < 0.001$), but not between successive days of the experiment (H = 1.01; $p = 0.40$). CAT values ranged from 0.02 to 0.19 U mg$^{-1}$ FW for *O. basilicum* leaves 14 days after treatment and *S. officinalis* leaves 1 day after treatment. In general, *M. officinalis* leaves showed higher CAT values than *O. basilicum* and *S. officinalis* (Table 6).

**Table 6.** Catalase activity (U mg$^{-1}$ FW) within leaves of *Ocimum basilicum*, *Melissa officinalis* and *Salvia officinalis* after *Tetranychus urticae* infestation.

| Treatments [1] | Catalase Activity (CAT) [2] | | |
| --- | --- | --- | --- |
| | *O. basilicum* | *M. officinalis* | *S. officinalis* |
| Control | 0.05 ($\pm$0.02) aA | 0.13 ($\pm$0.02) aA | 0.18 ($\pm$0.04) aA |
| 1 day | 0.03 ($\pm$0.00) aB | 0.14 ($\pm$0.02) aAB | 0.19 ($\pm$0.02) aA |
| 7 days | 0.04 ($\pm$0.02) aA | 0.14 ($\pm$0.02) aA | 0.06 ($\pm$0.02) aA |
| 14 days | 0.02 ($\pm$0.00) aB | 0.17 ($\pm$0.02) aA | 0.05 ($\pm$0.00) aAB |

[1] Treatments are as follow: control (leaves of plants without mites). 1 day (leaves of plants after 1 day of infestation). 7 days (leaves of plants after 7 days of infestation) and 14 days (leaves of plants after 14 days of infestation). [2] Means ($\pm$SE) with the same letter within columns (lowercase letters) and within rows (uppercase letters) are not significantly different ($p \leq 0.05$; Kruskal–Wallis' test).

## 4. Discussion

Among spider mites, *T. urticae* is one of the most important plant pests with a wide host range and an extreme record of pesticide resistance [17,46,54]. Previous findings have indicated that phylogeny, biogeography and chemistry affect host plant selection by phytophagous species while host plant quality can affect many life-history traits of mite herbivore [55–59]. It was documented that *T. urticae* individual populations did not develop equally well during feeding on all potential plant hosts [46]. Our results in the behavioral test indicated lower preference of *T. urticae* toward *S. officinalis*, while *O. basilicum* plants were the most preferred by this mite species. The plant species of the highest degree of acceptance by *T. urticae* in free choice test, and the most abundantly colonized during two weeks of mite feeding was *O. basilicum*. Additionally, the highest percentage of the necrotic area on infested leaves as a measure to determine the damage cause by *T. urticae* was noted on *O. basilicum* and the lowest on *S. officinalis*. Among the studied medicinal plant species as a host for *T. urticae*, *S. officinalis* appeared to be the plant

of the lowest degree of susceptibility to two-spotted spider mite. The results of the choice and non-choice tests showed that sage plants were better defended than lemon balm and basil plants. Female adult longevity, oviposition period and life cycle duration were the shortest when mites reared on *O. basilicum* and the longest on *S. officinalis*. According to other researchers [60–62], shorter developmental times and higher reproduction of herbivore species on a host plant indicate the higher susceptibility of a host plant. It was demonstrated in numerous studies that the fecundity of spider mite females differs on various host plants [54,59,60,62–64]. Similarly, in this experiment, the highest total and daily fecundity was recorded on *O. basilicum* and the lowest on *S. officinalis*. These results suggest that the quantity and/or quality of nutritional contents of *O. basilicum* were more appropriate for two-spotted spider mites than the other species of tested medicinal plants. In addition, as reported by van Den Boom et al. [65] and Hoy [66] differences in fecundity could be related to host plant leaf morphology.

It has been observed that the presence of glandular and non-glandular trichomes, which act as physical barriers to small arthropod herbivores, is a major determinant of host acceptance by mites [67,68]. The observed differences in plant acceptance by mites were influenced by morphological and anatomical properties of the leaves. It has been documented in many studies and in the present research that sage leaves have many trichomes and additionally this plant produces high amounts of essential oils [69–71], while basil leaves are different in that respect [6]. Feeding of two-spotted spider mite on *S. officinalis* resulted in the reduction of its biological parameters in relation to the values obtained on *O. basilicium* and *M. officinalis*. Differences in the number of leaf trichomes noted in this experiment can affect host-plant acceptance and may explain the different number of eggs laid on various host plants in the present study. *S. officinalis*, which was characterized by the highest number of leaf trichomes, was the least accepted by spider mites, and *T. urticae* females feeding on sage had the lowest fertility. In contrast, Oku et al. [72] have showed host plant acceptance by *Tetranychus kanzawai* females on different plants was not included in this model. It was primarily affected by leaf hair traits, not by female fecundity (host quality). Moreover, plant trichomes produce large amounts of essential oil, substances from the group of secondary metabolites play the role of a defensive mechanism of a plant against the attack of various pathogens or herbivorous [67,73–75]. Presumably, in the present study, the composition of essential oils produced by the tested medicinal plants also influenced the acceptance and plant colonization by spider mites as well as their biological parameters. *S. officinalis*, which showed the lowest susceptibility to *T. urticae* feeding, was characterized by a high content of essential oils (1.32%) and strong substances with a toxic effect were found in their composition, e.g., thujone, cyneol, camphor, borneol and pinene. In contrast, Kowalski et al. [76] documented that *M. officinalis* leaves had lower levels (0.17%) of volatile substances compared to *S. officinalis*, while Juliani and Simon [77] have shown that the antioxidant activity of sweet basil ethanolic extract was very low. In this study, these plants were more likely to be colonized and to a greater extent damaged by mites. Studies by other authors [73,78,79] also proved the negative impact of sage oils on feeding and reduction of phytophagous demographic parameters. According to Won-Il et al. [80], sage oil applied to *T. urticae* caused more than 80% of adult mortality. However, *M. officinalis* antifeedant effect was also demonstrated against *Tribolium castaneum* [74], while *O. basilicum* showed a contact and fumigant toxicity against stored-food mite *Tyrophagus putrescentiae* [81].

According to the literature, the chemical composition may play an important role in plant physiological processes, such as protection against environmental stresses (e.g., herbivory infection), signal molecules in plant-pathogen interactions, structural constituents of cell walls, and it may be the main source of sage plant resistance against herbivorous species [82–84]. In this study, during all time points of the experiment, near fifty percent of mites accepted *O. basilicum* while *S. officinalis* plants was the least preferred. However, despite the unfavorable properties of *S. officinalis* for mites, two-spotted spider mite could



adapt to marginal hosts, which means that they evolved mechanisms that can overcome initially effective plant defenses [46].

Plants rely on a set of mechanisms to detect herbivores in order to respond with specific defense mechanisms. Previous findings indicated that the effect of mite attack correlated with changes in the plant antioxidant status [16]. Early events in plant-herbivore interactions commence with membrane potential depolarization at the feeding site, alteration in cell membrane and ion imbalance, followed by changes in intracellular Ca2+ and reactive oxygen and/or nitrogen species (ROS/RNS) generation [17]. The production of reactive oxygen species (ROS) in plant tissues is a rapid response to herbivore attack [85]. $H_2O_2$ produced in peroxisomes is usually quickly decomposed by catalase [86]. On the one hand, $H_2O_2$ is a signaling molecule, and on the other it is toxic to cells [87]. Previous findings indicated that insect feeding [88] and infestation with *T. urticae* [47] increased $H_2O_2$ content in plant tissues, which according to the authors was related to oxidative damage caused by arthropod infestation. However, despite its importance, the role of ROS-metabolizing systems in mite-induced plant responses is unknown [17].

Our research also showed higher values of hydrogen peroxide concentration in all herbal species compared to control, but these changes were not significant. According to Santamaria et al. [16] the role of $H_2O_2$ in signal transduction against spider mites remains complex and the interpretation of mite-plant ROS signals should include ROS-related genes as well as additional variables, such as enzymatic production/elimination of ROS or their compartmentation. It was demonstrated that $H_2O_2$ and electrolytic leakage levels increased after mite infestation. This response was associated with plant susceptibility to mite herbivory. In contrast, the infestation of mite over-expressing plants lead a moderate accumulation of $H_2O_2$ without changes in cellular electrolytic leakage, which was associated with a higher resistance to mite herbivory [16].

Malondialdehyde plays a key role as an indicator of cellular free-radical generation, thus it is involved in plant defense signaling [89]. MDA content in our study was significantly increased after 7 days of stress induced by *T. urticae* feeding on *O. basilicum* leaves or after 14 days on *M. officinalis* and *S. officinalis* leaves. This suggests that antioxidant defense systems in all herbal plants were activated in longer periods of infestation. Our finding is consistent with reports that feeding of phloem-sucking insects increases lipid peroxidation [31,34,90]. Responses to oxidative stress were manifested in the maximum expression of antioxidant enzymes. Peroxidases and catalase act as ROS-scavenging enzymes that reduce ROS accumulation and detoxify oxidation products, thus their activity indicates that stress has been induced [31,91,92]. Their role in defense mechanisms against insect [43,93] and mite [47] feeding has been observed in various plant species. In our study, higher GPX activity was observed in all experimental variants. However, a significant increase, compared to control, was noted after two weeks of *T. urticae* feeding on *O. basilicum*, after one week on *M. officinalis* and after 1 day on *S. officinalis*. Long-term feeding of two-spotted mites on *M. officinalis* and *S. officinalis* decreased GPX activity, as in the case of scale insect [33] and aphid [31] feeding.

The present study showed that mite feeding insignificantly down-regulated catalase activity in *O. basilicum* and *S. officinalis* leaves, while slightly increased activity of this enzyme was observed in infested leaves of *M. officinalis*. A decrease in CAT activity was also found in bean plants in response to spider mite feeding [47] as well as in wheat plants colonized by Russian wheat aphids [93].

Additionally, *S. officinalis* leaves contained elevated levels of hydrogen peroxide and malondialdehyde compared to control; in addition, higher activity of GPX and lower catalase activity were recorded with an increased duration of infestation. As a result, the number of *T. urticae* and the percentage of damage to *S. officinalis* leaf tissues observed after two weeks were the lowest compared to *O. basilicium* and *M. officinalis*. In turn, *T. urticae* population on *O. basilicum* was characterized by the highest values of the analyzed biological parameters. *O. basilicum* was the most infested host for two-spotted spider mites due to the highest level of acceptance, increasing levels of hydrogen peroxide and MDA

with elevated GPX activity and strongly decreased catalase activity during all time points of mite infestation. This plant also had the largest mite population and the highest percentage of leaf damage after two weeks of *T. urticae* feeding.

## 5. Conclusions

We can state that plant-mite interaction differs depending on the plant species and affects their biological parameters. The investigated medicinal plants varied in their degree of spider mite preference. Among them, *S. officinalis* was the least preferred plant with the lowest degree of susceptibility to two-spotted spider mites feeding compared to *M. officinalis* and *O. basilicum*, while basil was the most acceptable one. This was evidenced by the lowest number of mites inhabiting *S. officinalis* and the level of leaf damage compared to *O. basilicum* and *M. officinalis*, as well as unfavorable values of biological parameters of *T. urticae* feeding on sage plant, causing its slow development and the values of the analyzed physiological parameters, as an indicator of the degree of plant direct defense. *T. urticae* infestation induced various levels of oxidative stress depending on the plant species and duration of mite infestation.

These results can be considered highly valuable in the use of trap plants in crops for pest control or the composition of mite repellents containing essential oils, allowing to reduce the amount of pesticides. The results presented here provide complementary knowledge on resistant medicinal plants that may be useful in predicting spider mite population growth. It can contribute in the future to strengthening integrated pest management programs for this mite in medicinal plant pot breeding.

Knowledge about plant-mite interactions and plant physiological responses to mites is very important for breeding strategies, especially in the case of medicinal plants and their health properties for humans. We currently continue our investigation on the medicinal plant responses to mites mainly the degree in which different plant species release spider mite induced volatiles and use it as an indirect defense to attract predatory mites. The defense system of higher plants is very complex and still requires further research, therefore the investigation of other means of control, using more agroecological methods as well as breeding strategies, may represent an attractive opportunity to managing mite-pests on medicinal plants in a way that is less aggressive to the human health and environment.

**Author Contributions:** Conceptualization, K.G. and I.G.J.; methodology, K.G., I.K. and E.G.-D.; software, M.K. and A.J.; formal analysis, K.G., I.G.J. and K.K.; investigation, K.G., E.G.-D.; resources, K.G. and I.G.J., data curation, K.G.; writing—original draft preparation, K.G.; writing—review and editing, I.G.J., B.Ł.; visualization, K.G., I.G.J.; project administration, B.S.-B; funding acquisition, K.G. All authors have read and agreed to the published version of the manuscript.

**Funding:** This research was funded by University of Life Sciences in Lublin, grant number OKE/DS/2 in 2013-2017.

**Institutional Review Board Statement:** Not applicable.

**Informed Consent Statement:** Not applicable.

**Data Availability Statement:** Not applicable.

**Acknowledgments:** Authors would like to thank heartily Katarzyna Rubinowska and Magdalena Michońska from Department of Botany and Plant Physiology, University of Life Sciences in Lublin for the performing physiological and botanical analyzes.

**Conflicts of Interest:** The authors declare no conflict of interest.

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
