# Peer review of "Defense Responses in the Interactions between Medicinal Plants from Lamiaceae Family and the Two-Spotted Spider Mite Tetranychus urticae Koch (Acari: Tetranychidae)"

_agronomy, doi:10.3390/agronomy11030438_

Round 1

Reviewer 1 Report

Line 213 - Kindly clarify this or is this an error 950C?

Line 222 - Ethylene instead of Ethyleno

Line 223-224 -  4oC change to proper superscript and check the rest of the article

Line 236 (and the rest of the article)- For H2O2, observed the proper subscript for this compound and check the rest of the article.

Line 259 -  T. urticae and other plant and insect species, kindly italicize all throughout the article

Line 355-356 : Reconstruct sentence, some grammatical errors.

Line 363 - percentage instead of percent

Line 382: host plant leaf instead of leave

Author Response

Date: 18.02.2020

To the Editor in Chief , Agronomy

Subject: Response to comments raised by REVIEWER 1

Ref.: Ms. ID agronomy-1106067

Manuscript title: "Defense responses in the interactions between medicinal plants from Lamiaceae family and the two-spotted spider mite Tetranychus urticae Koch (Acari: Tetranychidae)" 

Dear Sir/Madam,

While thanking you for very valuable comments and suggestions made on the MS, I would like to submit the revised Ms along with responses to reviewers’ comments which are listed below for your kind attention.  We highlight the changes to our manuscript within the document by using the track changes mode in MS Word.

 Line 213 - Kindly clarify this or is this an error 950C?  This has been corrected. Probably these error appeared during the edition of ms

Line 222 - Ethylene instead of Ethyleno: It was corrected

Line 223-224 -  4oC change to proper superscript and check the rest of the article: This has been carefully corrected. Probably these error appeared during the edition of ms

Line 236 (and the rest of the article)- For H2O2, observed the proper subscript for this compound and check the rest of the article: this has been carefully corrected in the whole text of submitted manuscript

Line 259 -  T. urticae and other plant and insect species, kindly italicize all throughout the article: this has been corrected in the whole text of submitted manuscript, as advised.

Line 355-356 : Reconstruct sentence, some grammatical errors. „Previous findings have indicated that phylogeny, biogeography and chemistry affect host plant selection by phytophagous species while host plant quality can affect of development their mite herbivore by impairing its life parameters”  We are agree with reviewer, this sentence has been change into: Previous findings have indicated that phylogeny, biogeography and chemistry affect host plant selection by phytophagous species while host plant quality can affect many life‐history traits of mite herbivore

Line 363 - percentage instead of percent: this has been corrected

Line 382: host plant leaf instead of leave: it has been corrected in the Discussion section

All these comments have been included in this revised submission.

Yours sincerely

Katarzyna Golan

Reviewer 2 Report

Dear editor,

In their article “Defense responses in the interactions between medicinal plants from Lamiaceae family and the two-spotted spider mite Tetranychus urticae Koch (Acari: Tetranychidae)” Golan et al. report that T. urticae infestation induces various levels of oxidative stress in some important medical crop plants; this pattern depends on the plant species and duration of infestation by spider mites. In turn, host plant species affected T. urticae life cycle parameters. For example, Salvia officinalis was the least infected plant and reduced mites demographic parameters. Infested leaves had increased levels of hydrogen peroxide and malondialdehyde compared to uninfested control leaves. In contrast, Ocimum basilicum was a suitable host on which T. urticae could develop and expand in number which seems to correlate with increasing levels of hydrogen peroxide and malondialdehyde as well as elevated guaiacol peroxidase activity, but decreased catalase activity.

Authors suggest applying their findings, i.e. differences in mite susceptibility of the tested medicinal plants, in breeding strategies and integrated T. urticae pest management in medicinal plant cultivations. Indeed, given that, among spider mites, T. urticae is one of the most important plant pests with a wide host range and high levels of pesticide resistance, this study reveals potential strategies how to avoid T. urticae infestion of important crop plants in a sustainable way. However, there are some important issues (e.g. presentation of the findings, references) to clarify before I can recommend publication in Agronomy. Please see below for details.

Issues:

- Introduction is too long: I suggest shortening it, e.g. deleting lines 71-78line 101: “H2O2 into N20 and O2” à numbers in chemical compounds, please write subscript throughout the whole manuscript!

- lines 134, 153 and 195: I am wondering whether it is important to mention in what department the experiments have been carried out? If so, why has this not been mentioned for all analyses?

- lines 263 and 266 “Figure 1” in is missing in the manuscript! Or do authors mean “Table 1”?

- line 274: Table 1: Why is “Ocimum basilicum L.” in bold?

- line 368-69: why full species name written here? Mention 1x at beginning only

- line 370: “were the shortest were mites were reared” à somehow wrong

- line 399-407: I am missing references relating to the importance of essential oils.

- line 382: start new paragraph after “host plant leave morphology.”

- line 418: since a new paragraph is started, it is unclear to what “this” in “…this chemical composition” refers to.

- line 445: “herb species” à correct to “herbal” species

- line 521: Reference list is messed up à in the text authors mentioned in the references, but annexed is a numbered list (where ref #1 is even missing!)

- Why not making graphs out some of the tables to improve clarity! I suggest making graphs (figures) out of Table 1 and 2 to further improve readability. Moreover, please put all standard errors values (±SE) in all tables into brackets!

- Table 3,4,5: “2Means within columns (lowercase letters) and within rows (uppercase letters) for each enzyme with the same letter are not significantly different (p<0.05) according to the Kruskal-Wallis test.” à really hard to understand, please improve!

- Write latin names in italics, throughout whole manuscript

Author Response

Date: 18.02.2020

To the Editor in Chief , Agronomy

Subject: Response to comments raised by REVIEWER 2

Ref.: Ms. ID agronomy-1106067

Manuscript title: "Defense responses in the interactions between medicinal plants from Lamiaceae family and the two-spotted spider mite Tetranychus urticae Koch (Acari: Tetranychidae)"

 Dear Sir/Madam,

While thanking you for very valuable comments and suggestions made on the MS, I would like to submit the revised Ms along with responses to reviewers’ comments which are listed below for your kind attention.  We highlight the changes to our manuscript within the document by using the track changes mode in MS Word.

- Introduction is too long: I suggest shortening it, e.g. deleting lines 71-78: Yes, we are agree with the reviewer that the section Introduction is too long. However, we decided to delete others two parts of  this section. In our opinion, lines 71-78 describe important information about physical plant properties which are important information connecting with our study. We decided to delete lines 41-43 and  63-70.

line 101: “H2O2 into N20 and O2” à numbers in chemical compounds, please write subscript throughout the whole manuscript! : This has been carefully corrected. Probably these error appeared during the edition of ms

- lines 134, 153 and 195: I am wondering whether it is important to mention in what department the experiments have been carried out? If so, why has this not been mentioned for all analyses? We are agree with the reviewer, this has been corrected and we decided to delete information about name of departments the experiments have been carried out. This information is not so important and it has no influence on the results of the conducted research. Thank you for this comment.

- lines 263 and 266 “Figure 1” in is missing in the manuscript! Or do authors mean “Table 1”? We are agree with the reviewer, we made a mistake,  and wrote the Figure 1 instesd Table 1; this has been corrected in the submitted manuscript

- line 274: Table 1: Why is “Ocimum basilicum L.” in bold? This has been corrected. Probably these error appeared during the edition of ms

- line 368-69: why full species name written here? Mention 1x at beginning only; We are agree with the reviewer, we made a mistake, this has been corrected, the full names were deleted from this part of Disscusion section

- line 370: “were the shortest were mites were reared” à somehow wrong: We are agree with the reviewer. This has been corrected ino: Females adult longevity, oviposition period and life cycle duration were the shortest for mites reared on O. basilicum and the longest on S. officinalis.

- line 399-407: I am missing references relating to the importance of essential oils. Yes, we are agree with the reviewer, it was corrected and references about essential oils have been added (Kostić et al., 2007; Glas et al., 2012; Ebadollahi et al., 2016; 2020)

- line 382: start new paragraph after “host plant leave morphology.” it has been corrected, as advised

- line 418: since a new paragraph is started, it is unclear to what “this” in “…this chemical composition” refers to. We are agree with the reviewer, this has been corrected. The word “this” was delete from this sentence  and change ino “the”.

- line 445: “herb species” à correct to “herbal” species: Of course, this has been corrected

- line 521: Reference list is messed up à in the text authors mentioned in the references, but annexed is a numbered list (where ref #1 is even missing!): We are agree with the reviewer, this section has been carefully corrected. 

- Why not making graphs out some of the tables to improve clarity! I suggest making graphs (figures) out of Table 1 and 2 to further improve readability. Moreover, please put all standard errors values (±SE) in all tables into brackets!  Thank you for your opinion but we don’t agree with suggestion about making graphs out of table 1 and 2. We tried to change the tables to graphs, but due to different parameters compared, it is difficult to put them together. We carefully re-editing all tables to improve their clarity, and putting all standard errors values (±SE) in tables into brackets,  as advised.

- Table 3,4,5: “2Means within columns (lowercase letters) and within rows (uppercase letters) for each enzyme with the same letter are not significantly different (p<0.05) according to the Kruskal-Wallis test.” à really hard to understand, please improve! This sentence was change into: 2Means (±SE) with the same letter within columns (lowercase letters) and within rows (uppercase letters) are not significantly different (P ≤ 0.05; Kruskal–Wallis’ test).

- Write latin names in italics, throughout whole manuscript: This has been corrected in the whole text of submitted manuscript, as advised.

 All comments and suggestions have been included in this revised submission.

Yours sincerely

Katarzyna Golan

Reviewer 3 Report

There are several in the reference list:

  • 11. "Carvalho et al., 2015" is on the reference list but is not on the text. Same for the author "Diaz-Riquelme et al, 2016" (20) and "Ebadollahi et al., 2020" (22). The same is for the authors listed under the numbers 27, 45, 51, 56, 62, 66, 69, 70, 78, 81, 82, 90, 93, 100, 101, 102 in the reference list.
  • Line 73: One reference is missing from the reference list (Dąbrowski, 1988)
  • Reference number 92 (Shoorooei et al., 2018): the year on the text is 2012 but on the reference list it's 2018.
  • Reference number 34 (Guo et al., 2014): the year on the text is 2006 but on the reference list it's 2014.
  • Reference number 43 (Jena S & Choudhuri MA, 1981): the surname on the text is spelled as Houdhuri (line 199).

The introduction of the article can be a little shorter as well.

Author Response

Date: 18.02.2020

To the Editor in Chief , Agronomy

Subject: Response to comments raised by REVIEWER 3

Ref.: Ms. ID agronomy-1106067

Manuscript title: "Defense responses in the interactions between medicinal plants from Lamiaceae family and the two-spotted spider mite Tetranychus urticae Koch (Acari: Tetranychidae)"

Dear Sir/Madam,

While thanking you for very valuable comments and suggestions made on the MS, I would like to submit the revised Ms along with responses to reviewers’ comments which are listed below for your kind attention.  We highlight the changes to our manuscript within the document by using the track changes mode in MS Word.

There are several in the reference list:11. "Carvalho et al., 2015" is on the reference list but is not on the text. Same for the author "Diaz-Riquelme et al, 2016" (20) and "Ebadollahi et al., 2020" (22). The same is for the authors listed under the numbers 27, 45, 51, 56, 62, 66, 69, 70, 78, 81, 82, 90, 93, 100, 101, 102 in the reference list: We are agree with reviewer, in this part of manuscript we made a lot of mistakes, we didn’t remove these references from list when we were change the text of manuscript. Now, this part has been carefully corrected in this revised submission

The list of correction:

Carvalho et al., 2015: this position was deleted from references

Diaz-Riquelme et al, 2016: this position was deleted from references

Ebadollahi et al., 2020: it was added into Discussion section

27: Glas et al. 2012: it was added into Disscussion section

45:  Kant et al. 2015: this position was deleted from references

51: Kiełkiewicz et al. 2009: this position was deleted from references

56: Koussevitzky et al. 2007: this position was deleted from references

62: Maheshwari et al. 2018: this position was deleted from references

66: Marinosci ert al. 2015: this position was added into Discussion section

69: Miresmaili et al. 2014: this position was deleted from references

70: Mittler 2002: this position was deleted from references

78: Pandey et al. 2014: this position was deleted from references

81: Phippen and Simon 1998: this position was deleted from references

82:  Phippen and Simon 2000: this position was deleted from references

90: Schoonhoven et al. 2005: this position was deleted from references

93: Simon et al. 1990: this position was deleted from references

100: Teklehaymanot and Giday 2007: this position was deleted from references

101: Trevisan et al. 2003: this position was deleted from references

102: Trotta et al. 2014: this position was deleted from references

  • Line 73: One reference is missing from the reference list (Dąbrowski, 1988): this position was added into revised ms: this reference was added into list of references
  • Reference number 92 (Shoorooei et al., 2018): the year on the text is 2012 but on the reference list it's 2018. We are agree with reviewer, in this part of manuscript we made a mistake, Now, this part has been corrected in this revised submission and the year of this reference was corrected into 2018 in the text of manuscript
  • Reference number 34 (Guo et al., 2014): the year on the text is 2006 but on the reference list it's 2014. We are agree with reviewer, the year of this reference was corrected into 2014 in the text of manuscript
  • Reference number 43 (Jena S & Choudhuri MA, 1981): the surname on the text is spelled as Houdhuri (line 199). We are agree, it was our mistake. It has been corrected in the text of manuscript.
  • The introduction of the article can be a little shorter as well: Yes, we are agree with the reviewer that the section Introduction is too long. We decided to delete lines 41-43 and 63-70 to make this section shorter.

All comments and suggestions have been included in this revised submission.

Yours sincerely

Katarzyna Golan

Reviewer 4 Report

Please find my suggestions for minor changes in the text highlighted in the MS.

Specific comments:

2.3. Choice test

I assume that the few bean leaves colonized by 150 females of T. urticae served as the source of mites for a very short period. I assume that the mite infested detached bean leaves quickly dried out and became unsuitable for sustaining mites. Please indicate for how long you observed living mites on these bean leaves. A week after the initiation of the choice test, no mites could colonize the test plants from these leaves, so I do not understand the logic of this extended period. If you test host acceptance by colonizing individuals, when assessing host acceptance, you should exclude those individuals that were produced on the host. Their number is influenced by the antibiotic effect of the host and not only by antixenosis. Please verify if you counted only adult females when you assessed host preference and excluded juveniles. I have a feeling that you did not want to estimate host selection of colonizing adults independently from the performance of these adults in the first two tests (choice and non-choice).

By the way, figure 1, which is reporting the results of this choice test, is missing from the MS.

2.5. Biological parameters of T. urticae

This test was initiated by using adult males and females of unknown age (randomly selecting adults from the stock colony). Based on your description in the materials and methods section, it seems to me, that you measured fecundity and longevity of these adults you picked from the stock culture. Since you report life cycle in table 2., which is the duration from egg to egg laying female, I assume, it suggests that you observed the development of the progeny (there are no details about this in materials and methods). If you measured adult life table parameters in the progeny, please indicate it and modify the materials and methods section. In its current form of the MS, it seems as if you report longevity and fecundity measured by using adults, whose emergence is unknown.

Results

In tables 1. and 2., what is the meaning of the last row? Please explain in the MS.

Author Response

 Date: 19.02.2020

 To the Editor in Chief , Agronomy

Subject: Response to comments raised by REVIEWER 4

 Ref.: Ms. ID agronomy-1106067

Manuscript title: "Defense responses in the interactions between medicinal plants from Lamiaceae family and the two-spotted spider mite Tetranychus urticae Koch (Acari: Tetranychidae)"

Dear Sir/Madam,

While thanking you for very valuable comments and suggestions made on the MS, I would like to submit the revised Ms along with responses to reviewers’ comments which are listed below for your kind attention.  We highlight the changes to our manuscript within the document by using the track changes mode in MS Word.

 Comments and Suggestions for Authors

Please find my suggestions for minor changes in the text highlighted in the MS:  all advised changes highlighted in the MS have been included in revised submission. Thank you for your comments.

Specific comments:

2.3. Choice test

I assume that the few bean leaves colonized by 150 females of T. urticae served as the source of mites for a very short period. I assume that the mite infested detached bean leaves quickly dried out and became unsuitable for sustaining mites. Please indicate for how long you observed living mites on these bean leaves. A week after the initiation of the choice test, no mites could colonize the test plants from these leaves, so I do not understand the logic of this extended period. If you test host acceptance by colonizing individuals, when assessing host acceptance, you should exclude those individuals that were produced on the host. Their number is influenced by the antibiotic effect of the host and not only by antixenosis. Please verify if you counted only adult females when you assessed host preference and excluded juveniles. I have a feeling that you did not want to estimate host selection of colonizing adults independently from the performance of these adults in the first two tests (choice and non-choice).

We agree with the reviewer, we made a mistake in the Material and methods section in the description of the choice test. This part of MS has been carefully corrected. The description of the choice test methodology has been changed in this submitted MS; this experiment refers only to the shortest - 24 hours observation period. The error remained after the corrections and changes introduced during writing the next version of this MS, because it has not been removed both in the methodology and in the results sections. Consequently, in the Results section the description was adjusted to the correct methodology and missing Figure 1 was inserted in the submitted ms.

By the way, figure 1, which is reporting the results of this choice test, is missing from the MS. This has been corrected in this revised submission. Figure 1 has been added in the result section.

2.5. Biological parameters of T. urticae

This test was initiated by using adult males and females of unknown age (randomly selecting adults from the stock colony). Based on your description in the materials and methods section, it seems to me, that you measured fecundity and longevity of these adults you picked from the stock culture. Since you report life cycle in table 2., which is the duration from egg to egg laying female, I assume, it suggests that you observed the development of the progeny (there are no details about this in materials and methods). If you measured adult life table parameters in the progeny, please indicate it and modify the materials and methods section. In its current form of the MS, it seems as if you report longevity and fecundity measured by using adults, whose emergence is unknown.

            Thank you for your comments, of course we have to agree with this. The text of this subsection  in Material and methods has been carefully completed in this revised submission.

Results

In tables 1. and 2., what is the meaning of the last row? Please explain in the MS. –The explanation has been added in the Material and methods sections, Statistical analysis (‘The results of the statistical analysis of the free-choice test and non-choice test to determine  a significant difference in analyzed means carried out with the Kruskal-Wallis' test reported with an H statistic, degrees of freedom and the P value are included in the table 1 and 2.’)

All comments and suggestions in the text highlighted in the MS have been included in this revised submission.

Yours sincerely

Katarzyna Golan
